# FARE: Provably Fair Representation Learning

**Nikola Jovanović, Mislav Balunović, Dimitar I. Dimitrov, Martin Vechev**
{nikola.jovanovic,mislav.balunovic,
dimitar.iliev.dimitrov,martin.vechev}@inf.ethz.ch
Department of Computer Science
ETH Zurich

## Abstract

Fair representation learning (FRL) is a popular class of methods that can replace the original dataset with a debiased synthetic one, which is then to be used to train fair classifiers. However, recent work has shown that prior methods achieve worse accuracy-fairness tradeoffs than originally suggested, dictating the need for FRL methods that provide provable bounds on unfairness of any downstream classifier, a challenge yet unsolved. In this work we address this challenge and propose Fairness with Restricted Encoders (FARE), the first FRL method with provable fairness guarantees. Our key insight is that restricting the representation space of the encoder enables us to derive fairness guarantees, while allowing empirical accuracy-fairness tradeoffs comparable to prior work. FARE instantiates this idea with a tree-based encoder, a choice motivated by advantages of decision trees when applied in our setting. Crucially, we develop and apply a practical statistical procedure that computes a high-confidence upper bound on the unfairness of any downstream classifier. In our experimental evaluation on several datasets we demonstrate that FARE produces tight upper bounds, often comparable with empirical results of prior methods, establishing the practical value of our approach.

## 1 Introduction

It has been repeatedly shown that machine learning systems deployed in real-world applications propagate training data biases, producing discriminatory predictions that can negatively affect population subgroups [1–7]. These observations have forced regulators into action, leading to directives [8, 9] which demand parties aiming to deploy such systems to ensure *fairness* [10] of their predictions. Mitigation of unfairness has become a key concern for organizations, with the highest increase in perceived relevance over the last year, out of all potential risks of artificial intelligence [11, 12].

**Synthetic data via fair representation learning**  A promising approach that attempts to address this issue is *fair representation learning* (FRL) [13–19]—a long line of work that preprocesses the data using an encoder $f$, transforming each datapoint $x \in \mathcal{X}$ into a debiased representation $z$. FRL can be viewed as a form of synthetic data generation that transforms input dataset into a new, debiased dataset. The key promise of FRL is that this debiased dataset can be given to other parties, who want to solve a prediction task without being aware of fairness (or potentially even being fine with discriminating), while ensuring that *any* downstream classifier they train on these representations has favorable fairness. However, recent work [20, 21, 16] has demonstrated that for some FRL methods it is possible to train significantly more unfair classifiers than originally claimed. This illuminates a major drawback of all existing work—their claim about fairness of the downstream classifiers holds only for the models they considered during the evaluation, and does not *guarantee* favorable fairness of other downstream classifiers trained on $z$. This is insufficient for critical applications where fairness is enforced by regulations, leading to our key question:

NeurIPS 2022 Workshop on Synthetic Data for Empowering ML Research.

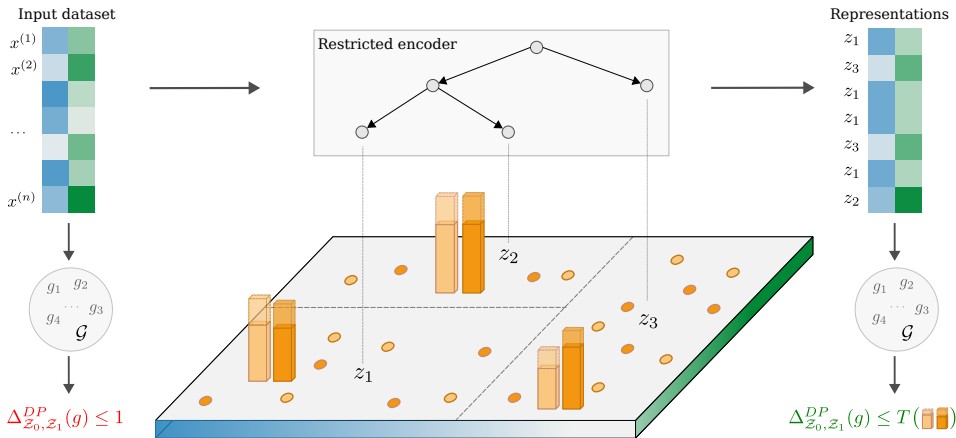

Figure 1: Overview of our provably fair representation learning method, FARE. The input dataset is transformed into fair representations using a restricted encoder. Our method can compute a provable upper bound $T$ on unfairness of any classifier $g \in \mathcal{G}$ trained on these representations.

*Can we create an FRL method that provably bounds the unfairness of any downstream classifier?*

The most prominent prior attempt to tackle this question, and the work most closely related to ours, is FNF [19]; we discuss other related work in Section 2. Assuming two groups $s = 0$ and $s = 1$ based on the sensitive attribute $s$, FNF shows that knowing the input distribution for each group can lead to an upper bound on unfairness of any downstream classifier. While this work is an important step towards provable fairness, the required assumption is unrealistic for most machine learning settings, and represents an obstacle to applying the approach in practice. Thus, the original problem of creating FRL methods that provide fairness guarantees remains largely unsolved.

**This work: provably fair representation learning**  We propose FARE (Fairness with Restricted Encoders, Fig. 1)—the first FRL method that offers provable upper bounds on the unfairness of any downstream classifier $g$ trained on its representations, without unrealistic prior assumptions. Our key insight is that using an encoder with *restricted representations*, i.e., limiting possible representations to a finite set $\{z_1, \ldots, z_k\}$, allows us to derive a practical statistical procedure that computes a high-confidence upper bound on the unfairness of any $g$, detailed in Section 4. FARE instantiates this idea with a suitable encoder based on fair decision trees, leading to a practical end-to-end FRL method which produces debiased representations augmented with strong fairness guarantees.

More concretely, FARE takes as input the set of samples $\{\boldsymbol{x}^{(1)}, \ldots, \boldsymbol{x}^{(n)}\}$ from the input distribution $\mathcal{X}$ (left in Fig. 1), and partitions the input space into $k$ *cells* (middle plane, $k = 3$ in this example) using the decision tree encoder. Finally, all samples from the same cell $i$ are transformed into the same representation $z_i$ (right). As usual in FRL, training a downstream classifier on representations leads to lower empirical unfairness, while slightly sacrificing accuracy on the prediction task.

However, the main advantage of FARE comes from the fact that using a restricted set of representations allows us to, using the given samples, estimate the distribution of two sensitive groups in each cell, i.e., compute an empirical estimate of the conditional probabilities $P(s = 0|\boldsymbol{z}_i)$ and $P(s = 1|\boldsymbol{z}_i)$ (solid color orange bars) for all $\boldsymbol{z}_i$. Further, we can use confidence intervals to obtain upper bounds on these values that hold with high probability (transparent bars). As noted above, this in turn leads to the key feature of our method: a tight upper bound $T$ on the unfairness of any $g \in \mathcal{G}$, where $\mathcal{G}$ is the set of all downstream classifiers that can be trained on the resulting representations. As we later elaborate on, increasing the number of samples $n$ makes the bounds tighter. Given the current trend of rapidly growing datasets, this further illustrates the practical value of FARE.

In our experimental evaluation in Section 5 we empirically demonstrate that on real datasets FARE produces tight upper bounds, i.e., the unfairness of any downstream classifier trained on FARE representations is tightly upper-bounded, which was not possible for any of the previously proposed FRL methods. Moreover, these downstream classifiers are able to achieve comparable empirical accuracy-fairness tradeoffs to methods from prior work. We believe this work represents a major

step towards solving the important problem of generating debiased synthetic data that provably prevents training of discriminatory machine learning models.

**Main contributions** The key contributions of our work are:

- A practical statistical procedure that, for restricted encoders, upper-bounds the unfairness of any downstream classifier trained on their representations.
- An end-to-end FRL method FARE, that instantiates this approach with a fair decision tree encoder, and applies the said statistical procedure to augment the synthetic dataset with a tight provable upper bound on unfairness of any downstream classifier.
- An extensive experimental evaluation in several settings, demonstrating favorable empirical fairness results, as well as tight upper bounds on unfairness (which were out of reach for prior work), often comparable to empirical results of existing FRL methods.

## 2 Related Work

We discuss related work on FRL, and prior attempts to obtain guarantees. See Appendix A for an additional discussion of literature on fair decision trees and provable fairness in other settings.

**FRL for group fairness** Following Zemel et al. [13] which originally introduced FRL, a plethora of different methods have been proposed based on optimization [22, 18], adversarial training [23, 24, 15, 25–29, 17], variational approaches [30, 14, 31, 32], disentanglement [33], mutual information [16, 34], and normalizing flows [19, 35]. No prior method restricts representations, which is a key step in our work. While Zemel et al. [13] map data to *prototypes*, this mapping is probabilistic, thus fundamentally incompatible with our bounding procedure (see Section 4).

**Towards fairness guarantees** The key issue is that most of these methods produce representations that have no provable guarantees of fairness. Concretely, this means that a machine learning model trained on the representations produced by these methods could have arbitrarily bad fairness. In fact, prior work [36, 20, 16] has shown that methods based on adversarial training often significantly overestimate the fairness of their representations. While some of them derive bounds on maximum possible unfairness [37, 16, 29], these are of purely theoretical nature and cannot be exactly computed in practice. Closest to our work is FNF [19] that can compute high-confidence bounds, but critically, assumes knowledge of the input probability distribution, which is rarely the case in practice. Our work makes no such assumption, which makes it significantly more practical.

## 3 Preliminaries

We now set up the notation and provide the background necessary to understand our contributions.

**Fair representation learning** Assume data $(\boldsymbol{x}, s) \in \mathbb{R}^d \times \{0, 1\}$ from a joint probability distribution $\mathcal{X}$, where each datapoint belongs to a group with respect to a sensitive attribute $s$. We focus on binary classification, i.e., given $y \in \{0, 1\}$ for each $\boldsymbol{x}$, we aim to build $g \colon \mathbb{R}^d \to \{0, 1\}$ to predict $y$ from $\boldsymbol{x}$. The goal is to maximize both accuracy and fairness of $g$ with respect to $s$, according to some definition. This often implies a slight accuracy loss, as these goals are generally at odds.

A large class of methods aims to directly produce $g$ with satisfactory fairness properties. A different group of methods, our focus here, preprocesses data by applying an encoder $f \colon \mathbb{R}^d \to \mathbb{R}^{d'}$ to obtain a new *representation* $\boldsymbol{z} = f(\boldsymbol{x}, s)$ of each datapoint, producing a debiased synthetic dataset. This induces a joint distribution $\mathcal{Z}$ of $(\boldsymbol{z}, s)$. The downstream classifier $g$ is now trained to predict $y$ from $\boldsymbol{z}$, i.e., now we have $g \colon \mathbb{R}^{d'} \to \{0, 1\}$. The main advantage of these methods is that by ensuring fairness properties of representations $\boldsymbol{z}$, we can limit the unfairness of *any* $g$ trained on data from $\mathcal{Z}$.

**Fairness metric** Let $\mathcal{Z}_0$ and $\mathcal{Z}_1$ denote conditional distributions of $\boldsymbol{z}$ where $s = 0$ and $s = 1$, respectively. In this work, we aim to minimize the *demographic parity distance* of $g$, reflecting the goal of equally likely assigning positive outcomes to inputs from both sensitive groups:

$$\Delta_{\mathcal{Z}_0, \mathcal{Z}_1}^{DP}(g) := |\mathbb{E}_{\boldsymbol{z} \sim \mathcal{Z}_0}[g(\boldsymbol{z})] - \mathbb{E}_{\boldsymbol{z} \sim \mathcal{Z}_1}[g(\boldsymbol{z})]|.$$

Our choice of metric is primarily motivated by consistency with prior work—other definitions (e.g., equalized odds) may be more suitable for a particular use-case [38], and our method can be easily adapted to support them, following the corresponding results of Madras et al. [15].

In the remainder of this work, we will use $p_0$ and $p_1$ to denote the PDFs of $\mathcal{Z}_0$ and $\mathcal{Z}_1$ respectively, i.e., $p_0(\boldsymbol{z}_i) = P(\boldsymbol{z}_i | s = 0)$ and $p_1(\boldsymbol{z}_i) = P(\boldsymbol{z}_i | s = 1)$ and $p$ to denote the PDF of the marginal distribution of $\boldsymbol{z}$. Similarly, we will use $q$ for the marginal distribution of $s$, and $q_i$ for the conditional distribution of $s$ for $\boldsymbol{z} = z_i$, i.e., $q_i(0) = P(s = 0 | \boldsymbol{z} = z_i)$ and $q_i(1) = P(s = 1 | \boldsymbol{z} = z_i)$.

## 4   FARE: Provable Fairness Bounds with Restricted Encoders

We present our key contributions, the derivation of provable unfairness bounds under the assumption of restricted encoders (explained shortly), and an instantiation based on decision trees.

**Optimal adversary**   Consider the adversary $h \colon \mathbb{R}^{d'} \to \{0, 1\}$ predicting group membership $s$, which aims to maximize the following balanced accuracy objective:

$$BA_{\mathcal{Z}_0, \mathcal{Z}_1}(h) := \frac{1}{2} \left( \mathbb{E}_{\boldsymbol{z} \sim \mathcal{Z}_0}[1 - h(\boldsymbol{z})] + \mathbb{E}_{\boldsymbol{z} \sim \mathcal{Z}_1}[h(\boldsymbol{z})] \right). \tag{1}$$

Let $h^\star$, such that for all $h$, $BA_{\mathcal{Z}_0, \mathcal{Z}_1}(h^\star) \geq BA_{\mathcal{Z}_0, \mathcal{Z}_1}(h)$, denote the *optimal adversary*. Intuitively, the optimal adversary predicts the group $s$ for which the likelihood of $\boldsymbol{z}$ under the corresponding distribution ($\mathcal{Z}_0$ or $\mathcal{Z}_1$) is larger. More formally, $h^\star(\boldsymbol{z}) = \mathbb{1}\{p_1(\boldsymbol{z}) \geq p_0(\boldsymbol{z})\}$, where $\mathbb{1}\{\phi\} = 1$ if $\phi$ holds, and 0 otherwise (see Balunović et al. [19] for a proof). As shown in Madras et al. [15],

$$\Delta_{\mathcal{Z}_0, \mathcal{Z}_1}^{DP}(g) \leq 2 \cdot BA_{\mathcal{Z}_0, \mathcal{Z}_1}(h^\star) - 1 \tag{2}$$

holds for any $g$, i.e., we can upper-bound the unfairness of any downstream classifier trained on data from $\mathcal{Z}$ by computing the balanced accuracy of the optimal adversary $h^\star$.

**Restricted encoders**   Prior work is unable to utilize Eq. (2) to obtain a fairness guarantee, as using unconstrained neural network encoders generally makes it intractable to compute the densities $p_0(\boldsymbol{z})$ and $p_1(\boldsymbol{z})$ that define the optimal adversary $h^*$. Notably, Balunović et al. [19] use normalizing flows, allowing computation of $p_0(\boldsymbol{z})$ and $p_1(\boldsymbol{z})$ under the assumption of knowing corresponding densities in the original distribution $\mathcal{X}$. In contrast, we propose a class of encoders for which we can derive a procedure that can upper-bound the RHS of Eq. (2), and thus the unfairness of $g$, without imposing any assumption in terms of knowledge of $\mathcal{X}$. We rely only on a set of samples $(\boldsymbol{z}, s) \sim \mathcal{Z}$, obtained by applying $f$ to samples $(\boldsymbol{x}, s) \sim \mathcal{X}$, readily available in the form of a given dataset.

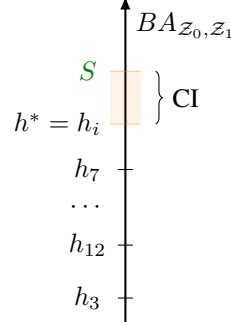

Namely, we hypothesize that restricting the space of representations can still lead to favorable fairness-accuracy tradeoffs. Based on this, we propose *restricted encoders* $f \colon \mathbb{R}^d \to \{\boldsymbol{z}_1, \ldots, \boldsymbol{z}_k\}$, i.e., encoders that map each $\boldsymbol{x}$ to one of $k$ possible values (*cells*) $\boldsymbol{z}_i \in \mathbb{R}^{d'}$. As now there is a finite number of possible values for a representation, we can use samples from $\mathcal{Z}$ to analyze the optimal adversary $h^*$ on each possible $\boldsymbol{z}$. More-

Figure 2: Restricted representations enable upper-bounding of $BA_{\mathcal{Z}_0, \mathcal{Z}_1}(h^\star)$.

over, we can upper-bound its balanced accuracy on the whole distribution $\mathcal{Z}$ with some value $S$ with high probability, using confidence intervals (CI) (as illustrated in Fig. 2). Finally, we can apply Eq. (2) to obtain the bound $\Delta_{\mathcal{Z}_0, \mathcal{Z}_1}^{DP}(g) \leq 2S - 1 = T$. A sketch of our upper-bounding procedure follows; see Appendix C for a detailed exposition.

**Upper-bounding the balanced accuracy**  We reformulate Eq. (1) as follows:

$$BA_{\mathcal{Z}_0,\mathcal{Z}_1}(h^*) = \frac{1}{2}\left(\sum_{i=1}^{k} p_0(\boldsymbol{z}_i)\cdot[1-h^*(\boldsymbol{z}_i)] + \sum_{i=1}^{k} p_1(\boldsymbol{z}_i)\cdot[h^*(\boldsymbol{z}_i)]\right) \qquad (\mathbb{E} \text{ of discrete RV})$$

$$= \frac{1}{2}\left(\sum_{i=1}^{k} \max\left(p_0(\boldsymbol{z}_i), p_1(\boldsymbol{z}_i)\right)\right) \qquad (\text{Optimal adversary})$$

$$= \sum_{i=1}^{k} p(\boldsymbol{z}_i)\cdot\max\left(\underbrace{(1/2q(0))}_{\alpha_0}\cdot q_i(0), \underbrace{(1/2q(1))}_{\alpha_1}\cdot q_i(1)\right), \qquad (\text{Bayes' rule})$$

where applications of Bayes' rule are $p_0(\boldsymbol{z}_i) = q_i(0)p(\boldsymbol{z}_i)/q(0)$ and $p_1(\boldsymbol{z}_i) = q_i(1)\cdot p(\boldsymbol{z}_i)/q(1)$. We do not know $\mathcal{Z}$, but instead have access a set $D$ of datapoints $(\boldsymbol{z}^{(j)}, s^{(j)}) \sim \mathcal{Z}$. Further, we assume a standard setting, where $D$ is split into a training set $D_{train}$, used to train $f$, validation set $D_{val}$, held-out for the upper-bounding procedure (and not used in training of $f$ in any capacity), and a test set $D_{test}$, used to evaluate the empirical accuracy and fairness of downstream classifiers.

Using these samples, we aim to obtain an upper bound $BA_{\mathcal{Z}_0,\mathcal{Z}_1}(h^*) \leq S$ that holds with confidence at least $1-\epsilon$, where $\epsilon$ is chosen in advance (we use $\epsilon = 0.05$). We heuristically choose a split $\epsilon = \epsilon_b + \epsilon_c + \epsilon_s$, and perform the upper-bounding procedure in three steps. First, we upper-bound the base rates $\alpha_0$ and $\alpha_1$ with confidence $1-\epsilon_b$, by applying the Clopper-Pearson binomial CI [39] (Appendix B) on $D_{train}$; this is sound as estimated probabilities are independent of the encoder $f$. Second, we use the obtained upper bounds to bound the per-cell balanced accuracy of $h^\star$, i.e., the expression $\max(\alpha_0 q_i(0), \alpha_1 q_i(1))$ for each cell $i$, with confidence $1-\epsilon_c$, again using Clopper-Pearson CI, this time on $D_{eval}$. Finally, we use the results of the previous steps to upper-bound the final sum with confidence $1-\epsilon_s$, applying Hoeffding's inequality [40] (Appendix B) on samples from $D_{test}$. Finally, we obtain the desired upper bound on the DP distance of any encoder $g$ trained on the embeddings from a restricted encoder:

$$\Delta^{DP}_{\mathcal{Z}_0,\mathcal{Z}_1}(g) \leq 2\cdot BA_{\mathcal{Z}_0,\mathcal{Z}_1}(h^\star) - 1 \leq 2S - 1 = T, \qquad (3)$$

which per union bound holds with desired error probability $\epsilon$, with respect to the sampling process.

This completes the bounding procedure, enabling provable fair representation learning with no restrictive assumptions. Our procedure can be applied to representations produced by any restricted encoder—here, we use a particular instantiation based on decision trees, that we describe next.

**Restricted representations with fair decision trees**  The restricted encoder used in FARE is based on decision trees, a choice motivated by strong results of tree-based models on tabular data [41], as well as their feature space splitting procedure, whose discrete behavior is inherently suitable for our requirement of restricted representations. In particular, we train a classification tree $f$ with $k$ leaves, and obtain a synthetic dataset by encoding all samples that end up in leaf $i$ to the same representation $\boldsymbol{z}_i$. We construct $\boldsymbol{z}_i$ based on the set of training examples in leaf $i$, taking the median value for continuous, and the most common value for categorical features (thus in our case, $d' = d$).

To be able to obtain good empirical results, and tight bounds, we modify the vanilla decision trees in two main ways. First, similar to Kamiran et al. [42] and others (see Appendix A), we generalize the Gini impurity criterion to optimize each split with respect to both fairness and accuracy (instead of just accuracy), introducing a tradeoff parameter $\gamma$. Second, we use ordinal encoding of categorical variables and generalize the usual techniques used in this case, again to make the procedure more fairness-aware. These changes are crucial to obtain a practical restricted encoder. We describe both changes in more detail and discuss the hyperparameters of FARE in Appendix D.

## 5   Experimental Evaluation

Here we evaluate FARE on several datasets, and show that its fairness-accuracy tradeoffs are comparable to prior work, while for the first time offering provable fairness bounds. We further investigate the tightness of our bounds and provide additional experiments on transfer learning in Appendix F.

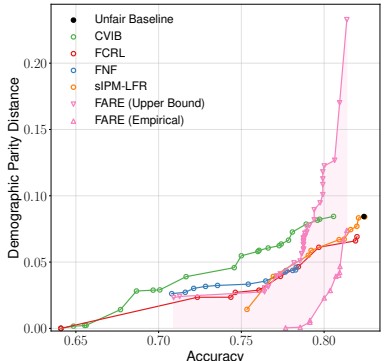 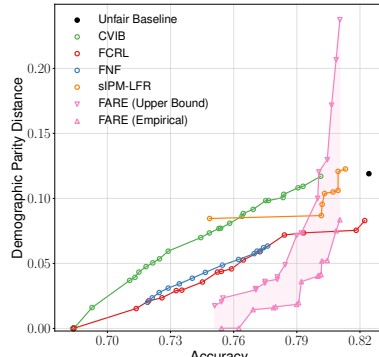

Figure 3: Evaluation of FRL methods on ACSIncome-CA (left) and ACSIncome-US (right).

**Experimental setup**  We consider common fairness datasets: Health [43] and two variants of AC-SIncome [44], ACSIncome-CA (only California), and ACSIncome-US (US-wide, larger but more difficult due to distribution shift). The sensitive attributes are age and sex, respectively. We compare our method with the following recent FRL baselines (described in Section 2): LAFTR [15], CVIB [14], FCRL [16], FNF [19], sIPM-LFR [17], and FairPath [18]. We provide all omitted details regarding datasets, baselines, and our experimental setup, in Appendix E.

**Main experiments**  We explore the fairness-accuracy tradeoff of each method by running it with various hyperparameters. Each run produces representations, used to train a 1-hidden-layer neural network (1-NN) for the prediction task using a standard training procedure (same for each method), and plot its demographic parity (DP) distance and prediction accuracy. Following Kim et al. [17], we show a test set Pareto front for each method. Further, for FARE we independently show a Pareto front of a 95% confidence provable upper bound on DP distance (following Section 4), which is a key feature of our approach and cannot be produced by any other method. Finally, we include an Unfair Baseline, which uses an identity encoder. The results on ACSIncome-CA and ACSIncome-US are shown in Fig. 3; the results on Health are given in Fig. 6 in Appendix F. We omit FairPath and LAFTR from the main plots (see extended results in Appendix F), as LAFTR has stability and convergence issues [16, 17], and FairPath uses a different metric to us [18].

Across all datasets, FARE can achieve a better or comparable accuracy-fairness tradeoff compared to baselines. Crucially, other methods cannot guarantee that there is no classifier with a worse DP distance when trained on their representations. This cannot happen for our method—we produce a *provable* upper bound on DP distance of *any* classifier trained on our representations. The results indicate that our provable upper bound is often comparable to *empirical* values of baselines. Finally, another advantage of FARE is its efficiency compared to the baselines (seconds instead of hours).

**Exploring downstream classifiers**  In Fig. 4, we show a representative point from our main experiments on Health (Fig. 6), its fairness guarantee, and 24 diverse downstream classifiers (see Appendix F) trained on same representations, where half are trained to maximize accuracy, and half to maximize unfairness. The latter (left cluster) can reach higher unfairness than initially suggested, reaffirming a known limitation of prior work [20, 16]: evaluating representations with some model class (here, a 1-NN) does not reliably estimate unfairness, as other classifiers (perhaps intentionally created by a malicious actor) might be more unfair. This highlights the value of FARE which provides a provable unfairness upper bound—all unfairness values still remain below a known upper bound. In Appendix F we perform the same analysis on another point from Fig. 3 (right), where since $k = 6$, it is possible to enumerate all $2^6 = 64$ possible classifiers, and directly confirm that each DP distance is below the upper bound.

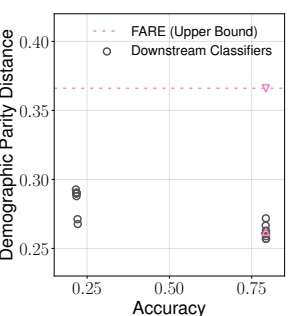

Figure 4: Comparing downstream classifiers with the FARE upper bound.

**Interpretability**  Finally, another advantage of FARE is that its tree-based encoder enables direct interpretation of representations. To illustrate this, for representations with $k = 6$ mentioned above we can easily find that, for example, the representation $z_6$ is assigned to each person older that $24$, with at least a Bachelor's degree, and an occupation in management, business or science.

## 6    Conclusion

We introduced FARE, a method to produce provably debiased synthetic data via fair representation learning. The key idea was that using restricted encoders enables a practical statistical procedure for computing a provable upper bound on unfairness of downstream classifiers trained on these representations. We instantiated this idea with a tree-based encoder, and experimentally demonstrated that FARE can for the first time obtain tight fairness bounds on several datasets, while simultaneously producing empirical fairness-accuracy tradeoffs similar to prior work which offers no guarantees.

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

## A  Additional Related Work

Here we discuss additional related work omitted from Section 2.

**Provable fairness in other settings**  Numerous related works on provable fairness provide a different kind of guarantee or assume a different setting than ours. First, in the setting of FRL, several methods have proposed approaches for learning individually fair representations [45–47], a different notion of fairness than group fairness which we focus on. Prior work has also examined provable fairness guarantees in various problem settings such as ranking [48], distribution shifting [49, 50], fair classification with in-processing [51, 52], individually fair classification with post-processing [53], and fair meta-learning [54]. These are all different from our setting, which is FRL for group fairness.

**Fair decision trees**  The line of work focusing on adapting decision trees to fairness concerns includes a wide range of methods which differ mainly in the branching criterion. Common choices include variations of Gini impurity [42, 55, 56], mixed-integer programming [57, 58] or AUC [59], while some apply adversarial training [60, 61]. Further, some works operate in a different setting such as online learning [56] or post-processing [62]. The only works in this area that offer provable fairness guarantees are Ranzato et al. [61], which certifies individual fairness for post-processing, and Meyer et al. [63], which certifies that predictions will not be affected by data changes. This fundamentally differs from our FRL setting where the goal is to certify fairness of any downstream classifier.

## B  Mathematical Tools

Here we formally state the Hoeffding's inequality and the Clopper-Pearson binomial confidence intervals, used in our upper-bounding procedure in Section 4.

*Hoeffding's inequality [40]*:  Let $X^{(1)}, \ldots, X^{(n)}$ be independent random variables such that $P(X^{(j)} \in [a^{(j)}, b^{(j)}]) = 1$. Let $\hat{\mu} = \frac{X^{(1)} + \ldots X^{(n)}}{n}$ and $\mu = \mathbb{E}[\hat{\mu}]$. It holds that:

$$P(\mu - \hat{\mu} \geq t) \leq \exp\left(\frac{-2n^2 t^2}{\sum_{i=1}^{n}(b^{(i)} - a^{(i)})^2}\right).$$

*Clopper-Pearson binomial proportion confidence intervals [39]*:  Assume a binomial distribution with an unknown success probability $\theta$. Given $m$ successes out of $n$ experiments, it holds that:

$$B(\frac{\alpha}{2}; m, n-m+1) < \theta < B(1 - \frac{\alpha}{2}; m+1, n-m) \tag{4}$$

with confidence at least $1 - \alpha$ over the sampling process, where $B(p; v, w)$ denotes the $p$-th quantile of a beta distribution with parameters $v$ and $w$.

## C  Detailed Description of the Upper-bounding Procedure

In this section we expand on the overview given in Section 4 and provide a detailed presentation of our practical statistical procedure used to upper-bound the unfairness of downstream classifiers trained on embeddings from a restricted encoder.

Recall from Section 4 that we aim to upper bound the following quantity with high probability, using samples from $\mathcal{Z}$:

$$BA_{\mathcal{Z}_0, \mathcal{Z}_1}(h^*) = \sum_{i=1}^{k} p(\boldsymbol{z}_i) \cdot \max\left(\underbrace{(1/2q(0))}_{\alpha_0} \cdot q_i(0), \underbrace{(1/2q(1))}_{\alpha_1} \cdot q_i(1)\right).$$

The expression above can be interpreted as the prior-weighted (i.e., weighted by $p(\boldsymbol{z}_i)$) per-cell balanced accuracy (i.e., $\max(\alpha_0 q_i(0), \alpha_1 q_i(1))$ for each cell $i$), where we define $\alpha_0 = 1/2q(0)$ and $\alpha_1 = 1/2q(1)$.

Next, we introduce 3 lemmas, and later combine them to obtain the desired upper bound. We use $B(p; v, w)$ to denote the $p$-th quantile of a beta distribution with parameters $v$ and $w$. Note that for Lemma 1 we do not use the values $\boldsymbol{z}^{(j)}$ in the proof, but still introduce them for consistency.

**Lemma 1** (Bounding base rates). *Given $n$ independent samples $\{(\boldsymbol{z}^{(1)}, s^{(1)}), \dots, (\boldsymbol{z}^{(n)}, s^{(n)})\} \sim \mathcal{Z}$ and a parameter $\epsilon_b$, for $\alpha_0$ and $\alpha_1$ as defined above, it holds that*

$$\alpha_0 < \frac{1}{2B(\frac{\epsilon_b}{2}; m, n - m + 1)}, \quad and \quad \alpha_1 < \frac{1}{2(1 - B(1 - \frac{\epsilon_b}{2}; m + 1, n - m))},$$

*with confidence $1 - \epsilon_b$, where $m = \sum_{j=1}^{n} \mathbb{1}\{s^{(j)} = 0\}$.*

*Proof.* We define $n$ independent Bernoulli random variables $X^{(j)} := \mathbb{1}\{s^{(j)} = 0\}$ with same unknown success probability $q(0)$. Using the Clopper-Pearson binomial CI [39] (Appendix B) to estimate the probability $q(0)$ we get $P(q(0) \leq B(\frac{\epsilon_b}{2}; m, n - m + 1)) \leq \epsilon_b/2$ and $P(q(0) \geq B(1 - \frac{\epsilon_b}{2}; m + 1, n - m)) \leq \epsilon_b/2$. Substituting $q(0) = 1 - q(1)$ in the latter, as well as the definitions of $\alpha_0$ and $\alpha_1$ in both inequalities, produces the inequalities from the lemma statement, which per union bound simultaneously hold with confidence $1 - \epsilon_b$. $\square$

**Lemma 2** (Bounding balanced accuracy for each cell). *Given $n$ independent samples $\{(\boldsymbol{z}^{(1)}, s^{(1)}), \dots, (\boldsymbol{z}^{(n)}, s^{(n)})\} \sim \mathcal{Z}$, parameter $\epsilon_c$, and constants $\bar{\alpha}_0$ and $\bar{\alpha}_1$ such that $\alpha_0 < \bar{\alpha}_0$ and $\alpha_1 < \bar{\alpha}_1$, it holds for each cell $i \in \{1, \dots, k\}$, with total confidence $1 - \epsilon_c$, that*

$$\max(\alpha_0 \cdot q_i(0), \alpha_1 \cdot q_i(1)) \leq t_i, \tag{5}$$

*where $t_i = \max\left(\bar{\alpha}_0 B(\frac{\epsilon_c}{2k}; m_i, n_i - m_i + 1), \bar{\alpha}_1 (1 - B(1 - \frac{\epsilon_c}{2k}; m_i + 1, n_i - m_i))\right)$. In this expression, $n_i = |Z_i|$, and $m_i = \sum_{j \in Z_i} \mathbb{1}\{s^{(j)} = 0\}$, where we denote $Z_i = |\{j | \boldsymbol{z}^{(j)} = \boldsymbol{z}_i\}|$.*

*Proof.* As in Lemma 1, for each cell we use the Clopper-Pearson CI to estimate $q_i(0)$ with samples indexed by $Z_i$ and confidence $1 - \epsilon_c/k$. As before, we apply $q_i(0) = 1 - q_i(1)$ to arrive at a set of $k$ inequalities of the form Eq. (5), which per union bound jointly hold with confidence $1 - \epsilon_c$. $\square$

**Lemma 3** (Bounding the sum). *Given $n$ independent samples $\{(\boldsymbol{z}^{(1)}, s^{(1)}), \dots, (\boldsymbol{z}^{(n)}, s^{(n)})\} \sim \mathcal{Z}$, where for each $j \in \{1, \dots, n\}$ we define a function $idx(\boldsymbol{z}^{(j)}) = i$ such that $\boldsymbol{z}^{(j)} = \boldsymbol{z}_i$ (cell index), parameter $\epsilon_s$, and a set of real-valued constants $\{t_1, \dots, t_k\}$, it holds that*

$$P\left(\sum_{i=1}^{k} p(\boldsymbol{z}_i) t_i \leq S\right) \geq 1 - \epsilon_s, \text{ where } S = \frac{1}{n} \sum_{j=1}^{n} t_{idx(\boldsymbol{z}^{(j)})} + (b - a)\sqrt{\frac{-\log \epsilon_s}{2n}}, \tag{6}$$

*and we denote $a = \min\{t_1, \dots, t_k\}$ and $b = \max\{t_1, \dots, t_k\}$.*

*Proof.* For each $j$ let $X^{(j)} := t_{idx(\boldsymbol{z}^{(j)})}$ denote a random variable. As for all $j$, $X^{(j)} \in [a, b]$ with probability 1 and $X^{(j)}$ are independent, we can apply Hoeffding's inequality [40] (restated in Appendix B) to upper-bound the difference between the population mean $\sum_{i=1}^{k} p(\boldsymbol{z}_i) t_i = \mathbb{E}_{\boldsymbol{z} \sim \mathcal{Z}} t_{idx(\boldsymbol{z})}$ and its empirical estimate $\frac{1}{n} \sum_{j=1}^{n} X^{(j)}$. Setting the upper bound such that the error is below $\epsilon_s$ directly recovers $S$ and the statement of the lemma. $\square$

**Applying the lemmas** Finally, we describe how we apply the lemmas in practice to upper-bound $BA_{\mathcal{Z}_0, \mathcal{Z}_1}(h^\star)$, and in turn upper-bound $\Delta_{\mathcal{Z}_0, \mathcal{Z}_1}^{DP}(g)$ for any downstream classifier $g$ trained on representations learned by a restricted encoder. We assume a standard setting, where a set $D$ of datapoints $\{(\boldsymbol{x}^{(j)}, s^{(j)})\}$ from $\mathcal{X}$ is split into a training set $D_{train}$, used to train $f$, validation set $D_{val}$, held-out for the upper-bounding procedure (and not used in training of $f$ in any capacity), and a test set $D_{test}$, used to evaluate the empirical accuracy and fairness of downstream classifiers.

After training the encoder and applying it to produce representations $(\boldsymbol{z}^{(j)}, s^{(j)}) \sim \mathcal{Z}$ for all three data subsets, we aim to derive an upper bound on $\Delta_{\mathcal{Z}_0, \mathcal{Z}_1}^{DP}(g)$ for any $g$, that holds with confidence at least $1 - \epsilon$, where $\epsilon$ is the hyperparameter of the procedure (we use $\epsilon = 0.05$). To this end, we heuristically choose some decomposition $\epsilon = \epsilon_b + \epsilon_c + \epsilon_s$, and apply Lemma 1 on $D_{train}$ to obtain upper bounds $\alpha_0 < \bar{\alpha}_0$ and $\alpha_1 < \bar{\alpha}_1$ with error probability $\epsilon_b$. As mentioned above, using $D_{train}$ in

this step is sound as estimated probabilities $q(0)$ and $q(1)$ are independent of the encoder $f$. Next, we use $\bar{\alpha}_0$, $\bar{\alpha}_1$ and $D_{val}$ in Lemma 2, to obtain upper bounds $t_1, \ldots, t_k$ on per-cell accuracy that jointly hold with error probability $\epsilon_c$. Finally, we upper-bound the sum $\sum_{i=1}^{k} p(\boldsymbol{z}_i)t_i \leq S$ with error probability $\epsilon_s$ using Lemma 3 on $D_{test}$ with previously computed $t_1, \ldots, t_k$. Combining this with Eq. (2) finally gives the desired upper bound, that per union bound holds with confidence $1 - \epsilon$:

$$\Delta_{\mathcal{Z}_0, \mathcal{Z}_1}^{DP}(g) \leq 2 \cdot BA_{\mathcal{Z}_0, \mathcal{Z}_1}(h^\star) - 1 \leq 2S - 1 = T. \tag{7}$$

# D   Fair Decision Trees as Restricted Encoders

Here we expand on our description of a decision tree as a restricted encoder given in Section 4. We start by recalling the background on decision trees and proceed to describe the two main additional components used in FARE.

**Vanilla classification trees**   Starting from the training set $D_{root}$ of examples $(\boldsymbol{x}, y) \in \mathbb{R}^d \times \{0, 1\}$, a binary classification tree $f$ repeatedly *splits* some leaf node $P$ with assigned $D_P$, i.e., picks a split feature $j \in \{1, \ldots, d\}$ and a split threshold $v$, and adds two nodes $L$ and $R$ as children of $P$, such that $D_L = \{(\boldsymbol{x}, y) \in D_P \mid x_j \leq v\}$ and $D_R = D_P \setminus D_L$. $j$ and $v$ are picked to minimize a chosen criterion, weighted by $|D_L|$ and $|D_R|$, aiming to produce leaves where the distribution of $y$ is highly unbalanced. We focus on Gini impurity, computed as $Gini_y(D) = 2p_y(1 - p_y) \in [0, 0.5]$ where $p_y = \sum_{(\boldsymbol{x}, y) \in D} \mathbb{1}\{y = 1\}/|D|$. At inference, a test example $\boldsymbol{x}$ is propagated to a leaf $l$, and we predict the majority class of $D_l$.

**Fairness-aware criterion**   Using a tree-based encoder that utilizes one of the common splitting criteria focused on accuracy (such as $Gini_y(D)$) generally leads to high unfairness, making it necessary to introduce a direct way to prioritize more fair tree structures. To this end, similar to Kamiran et al. [42] and others (see discussion in Appendix A), we use the criterion $FairGini(D) = (1 - \gamma)Gini_y(D) + \gamma(0.5 - Gini_s(D)) \in [0, 0.5]$, where $Gini_s$ is defined analogously to $Gini_y$. The second term aims to *maximize $Gini_s(D)$*, i.e., make the distribution of $s$ in each leaf $i$ as close to uniform (making it challenging for the adversary to infer the value of $s$ based on $\boldsymbol{z}_i$), while the hyperparameter $\gamma$ controls the accuracy-fairness tradeoff.

**Fairness-aware categorical splits**   Further, while usual splits of the form $x_j \leq v$ are suitable for continuous, they are inefficient for categorical (usually one-hot) variables, as only 1 category can be isolated. Consequently, this increases the number of cells and makes our fairness bounds loose. Instead, we represent $n_j$ categories for feature $j$ as integers $c \in \{1, 2, ..., n_j\}$. To avoid evaluating all $2^{n_j} - 1$ possible partitions, we sort the values by $p_y(c) = \sum_{(\boldsymbol{x}, y) \in D_c} \mathbb{1}\{y = 1\}/|D_c|$ where $D_c = \{\boldsymbol{x} \in D \mid x_j = c\}$, and consider all prefix-suffix partitions (*Breiman shortcut*).

This ordering focuses on accuracy and is provably optimal for $FairGini(D)$ with $\gamma = 0$ [64]. However, as it ignores fairness, it is inefficient for $\gamma > 0$. To alleviate this, we generalize the Breiman shortcut, and explore all prefix-suffix partitions under several orderings. Namely, for several values of the parameter $q$, we split the set of categories $\{1, 2, \ldots, n_j\}$ in $q$-quantiles with respect to $p_s(c)$ (defined analogous to $p_y(c)$), and sort each quantile by $p_y(c)$ as before, interspersing $q$ obtained arrays to obtain the final ordering. Note that while this offers no optimality guarantees, it is an efficient way to consider both objectives, complementing our fairness-aware criterion.

**Hyperparameters**   There are four main hyperparameters of FARE: $\gamma$ (used for the criterion, where larger $\gamma$ puts more focus on fairness), $\bar{k}$ (upper bound for the number of leaves), $\underline{n_i}$ (lower bound for the number of examples in a leaf), and $v$ (the ratio of the training set to be used as a validation set). Note that all parameters affect accuracy, empirical fairness, and the tightness of the fairness bound. For example, larger $\underline{n_i}$ is likely to improve the bound by making Lemma 2 tighter, as more samples can be used for estimation. For the same reason, increasing $v$ improves the tightness of the bound, but may slightly reduce the accuracy as fewer samples remain in the training set used to train the tree.

| Dataset | Training size | Test size | Base rate ($s$) | Base rate ($y$) |
|---|---|---|---|---|
| ACSIncome-CA | 165 546 | 18 395 | 0.46 | 0.64 |
| ACSIncome-US | 1 429 070 | 158 786 | 0.48 | 0.68 |
| Health | 174 732 | 43 683 | 0.35 | 0.68 |

Table 1: Statistics of evaluated datasets.

# E   Details of Main Experiments

In this section we provide details of our main experimental evaluation omitted from the main text.

**Datasets**   As mentioned in Section 5, we perform our experiments on ACSIncome [44] and Health [43] datasets. In Table 1 we show some general statistics about the datasets: size of the training and test set, base rate for the sensitive attribute $s$ (percentage of the majority group out of the total population), and base rate for the label $y$ (accuracy of the majority class predictor).

ACSIncome is a dataset recently proposed by Ding et al. [44] as an improved version of UCI Adult, with comprehensive data from US Census collected across all states and several years (we use 2014). The task is to predict whether an individual's income is above $50,000, and we consider sex as a sensitive attribute. We evaluate our method on two variants of the dataset: ACSIncome-CA, which contains only data from California, and ACSIncome-US, which merges data from all states and is thus significantly larger but also more difficult, due to distribution shift. 10% of the total dataset is used as the test set. We also use the Health dataset [43], where the goal is to predict the Charlson Comorbidity Index, and we consider age as a sensitive attribute (binarized by thresholding at 60 years). For this dataset perform the same preprocessing as Balunović et al. [19], and use 20% of the total dataset as the test set.

**Evaluation procedure**   For our main experiments, as a downstream classifier we use a 1-hidden-layer neural network with hidden layer size 50, trained until convergence on representations normalized such that their mean is approximately 0 and standard deviation approximately 1. We train the classifier 5 times and in our main figures report the average test set accuracy, and the maximal DP distance obtained, following the procedure of Gupta et al. [16].

**Hyperparameters**   For baselines, we follow the instructions in respective writeups, as well as Gupta et al. [16] to densely explore an appropriate parameter range for each value (linearly, or exponentially where appropriate), aiming to obtain different points on the accuracy-fairness curve. For CVIB, we explore $\lambda \in [0.01, 1]$ and $\beta \in [0.001, 0.1]$. For FCRL on ACSIncome we explore $\lambda = \beta \in [0.01, 2]$, and for Health $\lambda \in [0.01, 2]$ and $\beta = 0.5\lambda$. For FNF, we explore $\gamma \in [0, 1]$. For sIPM-LFR, we use $\lambda \in [0.0001, 1.0]$ and $\lambda_F \in [0.0001, 100.0]$, extending the suggested ranges. For FairPath we set the parameter $\kappa \in [0, 100]$. Finally, for LAFTR we use $g \in [0.1, 50]$, extending the range of $[0, 4]$ suggested by [16]. We adjust the parameters for transfer learning whenever supported by the method.

Regarding FARE (see description of hyperparameters in Appendix D), we investigate $\gamma \in [0, 1]$, $k \in [2, 200]$, $n_i \in [50, 1000]$, $v \in \{0.1, 0.2, 0.3, 0.5\}$. For the upper-bounding procedure, we always set $\epsilon = 0.05$, $\epsilon_b = \epsilon_s = 0.005$, and thus $\epsilon_c = 0.04$. Finally, when sorting categorical features as described in Appendix D, we use $q \in \{1, 2, 4\}$ in all cases.

# F   Additional Experimental Results

In this section we provide additional experimental results omitted from the main text. For main experiments, we provide results on the Health dataset and with two additional methods. Further, we provide an additional investigation of downstream classifiers, results of our experiment on the relationship of data and bound tightness, and a set of experiments on transfer learning.

**Main experiments**   In Fig. 5 and Fig. 6 we provide the extended results of our main experiments, including another dataset (Health), and two originally excluded methods, LAFTR and FairPath.

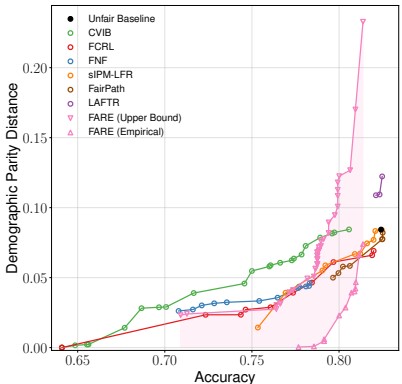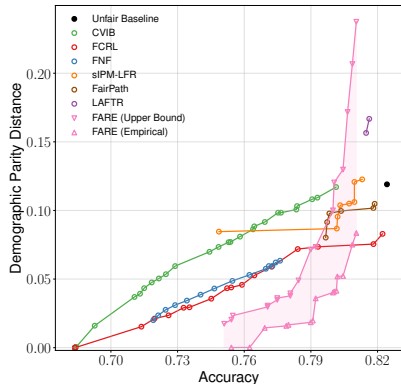

Figure 5: Extended evaluation on ACSIncome-CA (left) and ACSIncome-US (right).

**Exploring downstream classifiers** We provide additional info on the experiment with downstream classifiers given in Fig. 4, and repeat a similar experiment in a different setting.

Namely, for Fig. 4 we explored the following classifiers: (i) 1-hidden-layer neural network (1-NN) with hidden layer sizes 50 and 200, (ii) 2-NN with hidden layers of size $(50, 50)$, as well as $(200, 100)$, (iii) logistic regression, (iv) random forest classifier with 100 and 1000 estimators, (v) decision tree with 100 and an unlimited number of leaf nodes. We trained all these classifiers with a standardization preprocessing step as described above. We further trained one variant of 1-NN, 2-NN, random forest, and logistic regression, on unnormalized data. All described models were trained both to predict the task label $y$, and to maximize unfairness, i.e., predict $s$, leading to 24 evaluated models.

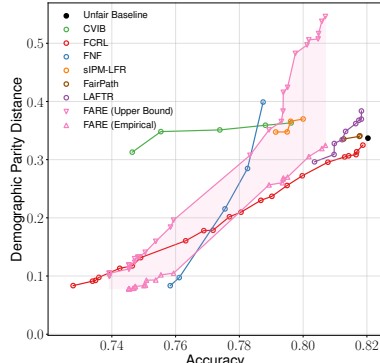

Figure 6: Main experimental evaluation on the Health dataset.

Next, in a similar vein, we explore a point from Fig. 3 (right), with accuracy 75.1% and DP distance of 0.005. As opposed to our previous experiment, here we have $k = 6$, i.e., the possible representations are $\{z_1, \ldots, z_6\}$, thus the previous investigation of downstream classifiers simplifies. Instead of choosing a model class, we can enumerate all $2^6 = 64$ possible classifiers, and directly confirm that each DP distance is below the upper bound, as shown in Fig. 7 (left). Note that in the original experiment, all baseline methods have DP distance $\geq 0.04$ at similar accuracy of $\approx 75\%$, implying that the FARE bound is in this case very tight.

**Data improves bounds** As mentioned in the main text, we investigate the effect of increased dataset size on bound tightness. Namely, we choose a representative set of FARE points from Fig. 3 (left), and repeat the upper-bounding procedure with the dataset repeated $M$ times, showing the resulting upper bounds for $M \in \{2, 4, 8, 16, 32\}$ in Fig. 7 (right). We can clearly observe a significant improvement in the provable upper bound for larger dataset sizes.

**Transfer learning** Finally, we analyze the transferability of learned representations across tasks. We produce a diverse set of representations on the Health dataset with each method, and following the procedure from prior work [15, 19, 17] evaluate them on five unseen tasks $y$, where for each the goal is to predict a certain primary condition group. For each task and each method, we identify the highest accuracy obtained while keeping $\Delta_{\mathcal{Z}_0, \mathcal{Z}_1}^{DP}$ not above a certain threshold. Moreover, we show $T$, the provable DP distance upper bound of FARE. The results are shown in Table 2. First, we observe that some methods are unable to reduce $\Delta_{\mathcal{Z}_0, \mathcal{Z}_1}^{DP}$ below the given threshold. Our method can always reduce the $\Delta_{\mathcal{Z}_0, \mathcal{Z}_1}^{DP}$ sufficiently, but due to our restriction on representations which enables provable upper bounds, we often lose more accuracy than other methods for high $\Delta_{\mathcal{Z}_0, \mathcal{Z}_1}^{DP}$ thresholds. Future work could focus on investigating alternative restricted encoders with better fairness-accuracy tradeoffs in the transfer learning setting.

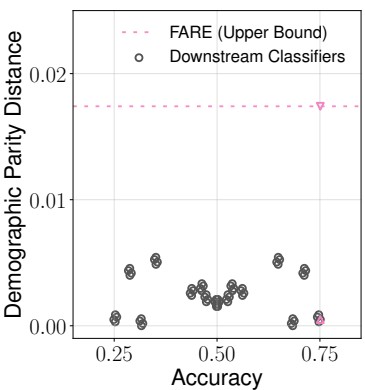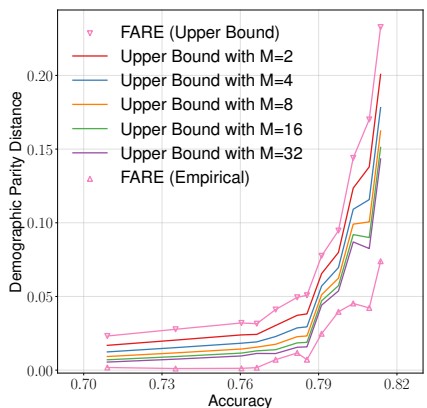

Figure 7: Comparing downstream classifiers with the FARE upper bound for a case where $k = 6$ (left). The impact of increasing the dataset size $M$ times on the fairness bound tightness (right).

| $y$ | $\Delta_{\mathcal{Z}_0,\mathcal{Z}_1}^{DP}$ | $T$ | FARE | FCRL | FNF | sIPM |
|---|---|---|---|---|---|---|
| MIS | $\leq 0.30$ | 0.64 | 79.3 | 78.6 | 79.2 | 79.8 |
| | $\leq 0.20$ | 0.64 | 79.3 | 78.6 | 78.9 | 79.8 |
| | $\leq 0.15$ | 0.64 | 79.3 | 78.6 | 78.9 | 79.6 |
| | $\leq 0.10$ | 0.48 | 78.8 | 78.6 | 78.9 | 79.0 |
| | $\leq 0.05$ | 0.54 | 78.7 | 78.6 | 78.7 | 78.6 |
| NEUMENT | $\leq 0.30$ | 0.64 | 73.2 | 72.4 | 71.9 | 78.8 |
| | $\leq 0.20$ | 0.64 | 73.2 | 72.4 | 71.9 | 76.6 |
| | $\leq 0.15$ | 0.64 | 73.2 | 72.4 | 71.8 | 73.2 |
| | $\leq 0.10$ | 0.64 | 73.2 | 72.2 | 71.8 | / |
| | $\leq 0.05$ | 0.42 | 72.1 | 71.4 | 71.7 | / |
| ARTHSPIN | $\leq 0.30$ | 0.41 | 74.4 | 70.7 | 68.9 | 78.3 |
| | $\leq 0.20$ | 0.41 | 74.4 | 70.7 | 68.9 | 78.3 |
| | $\leq 0.15$ | 0.46 | 74.2 | 70.1 | 68.9 | / |
| | $\leq 0.10$ | 0.23 | 69.5 | 69.6 | 68.7 | / |
| | $\leq 0.05$ | 0.23 | 69.5 | 69.5 | 68.5 | / |
| METAB3 | $\leq 0.30$ | 0.47 | 74.0 | 72.5 | 76.2 | / |
| | $\leq 0.20$ | 0.46 | 69.8 | 69.2 | 75.0 | / |
| | $\leq 0.15$ | 0.33 | 68.7 | 67.9 | 73.2 | / |
| | $\leq 0.10$ | 0.12 | 66.1 | 66.7 | 73.2 | / |
| | $\leq 0.05$ | 0.12 | 66.1 | 65.3 | / | / |
| MSC2a3 | $\leq 0.30$ | 0.56 | 71.3 | 70.5 | 73.5 | 77.6 |
| | $\leq 0.20$ | 0.53 | 67.2 | 70.5 | 73.0 | / |
| | $\leq 0.15$ | 0.12 | 63.0 | 69.7 | / | / |
| | $\leq 0.10$ | 0.12 | 63.0 | 69.0 | / | / |
| | $\leq 0.05$ | 0.12 | 63.0 | / | / | / |

Table 2: Results of transfer learning experiments on Health.

