# OpenReview forum: "FARE: Provably Fair Representation Learning"
_NeurIPS.cc/2022/Workshop/SyntheticData4ML — Neurips 2022 SyntheticData4ML_

### Official Review · Reviewer_KhM3 · 2022-10-09
**Important concerns about the novelty claim**

**Rating:** 5
**Confidence:** 4

**Review:**

The paper addresses a very significant topic and how the prior work is not sufficient to tackle the fairness problem.
It is well-written in terms of clarity. However, there are few points that need more explanation:
- The authors claim that their work is the first provable fairness method. Considering the 2017 paper called "Provably Fair Representations" that addressed this problem by information theoretically proving the fairness, how do the authors relate their paper to this prior work? Since they didn't even mention, cite or discuss this anywhere in the paper, their claim on proposing the first provable method seems unreliable. It's better to describe how their provable method is different or better (even quantitatively if possible and with reasoning if not possible to compare)
- The proposed method seems inspired by source coding which also inspired privacy (especially information theoretic privacy), however these are not clearly mentioned in the paper.
These points, mostly the first one, cause questioning the reliability of the paper's claim.

Experimental results of the paper are focused on binary sensitive groups and binary classification as the downstream task which covers a quite small application area, however the upper bound results are consistent and makes the point that the paper claims.

I am overall happy to increase my score after the mentioned concerns are addressed properly.

---

### Official Review · Reviewer_Te7M · 2022-10-10
**An interesting approach to group fairness**

**Rating:** 6
**Confidence:** 3

**Review:**

This work proposes the use of decision trees to embedding data into a discrete set of latent vectors. This approach distinguishes itself from other fairness approaches, e.g., optimization, adversarial training, variational approaches. This approach is reasonable: if the features are from a finite set, then the fairness should be bounded. In the extreme case, if the embedding is a constant, then the algorithm is absolutely fair. However, one shortcoming is that the upper bound can be quite loose when we requires the accuracy to be high, as can be seen from Figure 3. Therefore, the limitation of this work is that it only works for achieving moderately high accuracy.

Minor:
1) Line 109: $f: \mathbb{R}^d \times \{0, 1\} \to \mathbb{R}^{d'}$?
2) why does the upper bound grow much faster than other algorithms when we increase the accuracy?

---

### Official Review · Reviewer_1Sx5 · 2022-10-11
**Overall good paper, but unclear what the limitations are**

**Rating:** 6
**Confidence:** 4

**Review:**

Overall, I think it is a good paper: it is clearly written, well-motivated and the idea of regarding fair representation learning as a way to generate synthetic data works nicely due to a well-chosen method---i.e. the tree structure provides interpretable representations, such that publishing the synthetic data by itself is more sensible. I have the following three questions about limitations:

1. How easy would it actually be to extend FARE to other definitions, like EO? (in relation to the claim in line 119).

2. Is the focus on an optimal adversary too strict and necessary? E.g., in practice, one may be more interested in studying the bias in a classifier that is optimal or very good w.r.t. the downstream task. To me it seems that the representation’s expressiveness would need to be very limited if we want to limit the worst possible bias, yet for most models that we are interested the bias may be less bad.

3. In the experimental section there are just $k=6$ leaves. This seems like a very limited setting, such that one may wonder how useful the representation is for more difficult tasks. How well would it extend to larger and more difficult datasets, where one would require more expressive representations (e.g. large number of leaves)?

I’m happy to raise my score if the above points (especially 3.) are addressed.

---

### Meta-Review · Area_Chair_rEmK · 2022-10-20

**Recommendation:** Accept

**Review:**

The reviewers agree that the paper addresses an important problem and that it is well-presented, hence I recommend accepting it into the workshop. That being said, the authors should consider addressing the points raised by reviewer KhM3 in the camera-ready version.